# Accumulation of Microcystin from *Oscillatoria limnetica Lemmermann* and *Microcystis aeruginosa* (*Kützing*) in Two Leafy Green Vegetable Crop Plants *Lactuca sativa* L. and *Eruca sativa*

**DOI:** 10.3390/plants11131733

**Published:** 2022-06-29

**Authors:** Asmaa Bakr, Mashail Nasser Alzain, Nurah M. Alzamel, Naglaa Loutfy

**Affiliations:** 1Botany and Microbiology Department, Faculty of Science, Sohag University, Sohag 82524, Egypt; asmaabakr2011@science.sohag.edu.eg; 2Department of Biology, College of Sciences, Princess Nourah Bint Abdulrahman University, P.O. Box 84428, Riyadh 11671, Saudi Arabia; 3Department of Biology, College of Sciences and Humanities, Shaqra University, Shaqra 11961, Saudi Arabia; nalzamel@su.edu.sa; 4Botany and Microbiology Department, Faculty of Science, South Valley University, Qena 83523, Egypt; naglaa.hassan@sci.svu.edu.eg

**Keywords:** *Microcystis aeruginosa*, *Oscillatoria limnetica*, accumulation, microcystins, *Lactuca sativa* L., *Eruca sativa*

## Abstract

The use of contaminated water to irrigate crop plants poses a risk to human health from the bioaccumulation potential of microcystins (MCs) in the edible tissues of vegetable plants. The main objective of this study is to determine the concentration of total microcystins (MC-LR and MC-RR) in leafy green plants (*Lactuca sativa* L. *var. longifolia* and *Eruca sativa*) that have previously been irrigated with polluted water. Integrated water samples were collected by cleaned plastic bottles at a depth of about 30 cm from one of the sources of water used to irrigate agricultural lands for crop plants. At the same time, samples from plants were also collected because this water from the lake farm is used for the irrigation of surrounding vegetable plants such as *Lactuca sativa* L. *var. longifolia* and *Eruca sativa*. The dominant species of cyanobacteria in water samples are *Microcystis aeruginosa* (*Kützing*) and *Oscillatoria limnetica Lemmermann*, which were detected with an average cell count 2,300,000 and 450,000 cells/mL, respectively. These two dominant species in water produced two MCs variants (MC-LR, -RR) that were quantified by high-performance liquid chromatography (HPLC). Dissolve and particulate MCs were detected in the irrigation waters by HPLC with concentrations of 45.04–600 μg/L. MCs in the water samples exceeded the WHO safety limit (1 μg/L) of MC in drinking water. In addition, the total concentration of Microcystin in *Lactuca sativa* L. *var. longifolia* and *Eruca sativa* were 1044 and 1089 ng/g tissues, respectively. The estimated daily intake (EDI) of microcystins by a person (60 kg) consuming 300 g of fresh plants exceeded the total daily intake guidelines (0.04 μg kg^−1^ body weight) for human food consumption. According to the findings of this study, irrigation water and plants used for human consumption should be tested for the presence of MCs regularly through critical and regularly monitored programs to prevent the accumulation and transfer of such toxins through the food web.

## 1. Introduction

Due to the nutrient enrichment of the aquatic environment and climate change, cyanobacterial blooms have become a worldwide problem [1,2]. Cyanobacteria can produce a huge spectrum of harmful and destructive toxins [3]. Cyanotoxins are secondary metabolites that can be classified chemically as cyclic peptides, alkaloids, or lipopolysaccharides (LPS) and functionally as hepatotoxins, neurotoxins, or dermatotoxins [4]. Among the diverse cyanotoxins types, microcystin (MCs) is the most frequent occurrence and is hazardous in the aquatic environment [5]. *Anabaena*, *Fischerella*, *Gloeotrichia*, *Nodularia*, *Nostoc*, *Oscillatoria*, *Microcystis aeruginosa*, and *Planktothrix* are the most common producers that produce extremely water-soluble and non-volatile MCs [6,7]. They are known as cyclic heptapeptides (D-Ala1-X2-D-MeAsp3-Y4-Adda-Arg5-D-Glu6-Mdha7). X and Ya are two L-amino acids found in the peptide ring. Over 100 MCs variants have been discovered and named based on the variable amino acids that complete their structures [8,9]. The World Health Organization (WHO) established a tolerable daily intake (TDI) for humans of 1 µg/L for MC-LR in drinking water and 0.04 µg/kg of body weight/day as a provisional guideline limit [10].

Hepatotoxins are poisonous to the liver and can cause liver damage that is due to the mechanism that they pick up from the intestine, blood, or the liver [11,12]. MC causes liver injury through the inhibition of protein phosphatase, which may lead successively to the accumulation of phosphorylated proteins in the liver, cell necrosis, massive hemorrhage, and death [13]. Furthermore, liver toxins may cause tumor promotion in both the liver and colon [14]. When water contaminated by MCs is used for agriculture, applied by either direct spraying or taking them through the roots, hepatotoxins were accumulated in plants, causing a hazardous risk [15]. Toxins may affect crops through their effect on either germination or growth rates of some vegetables, such as lettuce or grasses [16]. In vitro studies have revealed that MC-LR may cause harm to other vital organs such as the kidney [17], thymus [18], and male reproductive organs [19,20]. Humans might be exposed to MC-LR by feeding crops irrigated with MC-contaminated water. Cyanotoxin accumulation in edible storage organs of agricultural crops such as stems, leaves, fruits, seeds, and corms has been documented using both direct irrigation and spray irrigation [21,22].

*Eruca sativa* Mill (commonly known as rocket salad and *Lactuca sativa*) are ancient crops of great economic and agronomic importance. They are a member of the Brassicaceae family. They contain vitamin C, as well as antioxidants such as phenolic compounds, carotenoids, and glucosinolates and degradation products such as isothiocyanates [23,24]. As a result, using cyanobacteria-contaminated water for irrigation may have an impact on the growth of these crops when they accumulate MCs in their edible tissues, posing a risk to human health if contaminated tissues are consumed. Many studies are reporting the effects of various concentrations of microcystin, both on the development of some plants and the accumulation of leaf tissue [25,26]. Ref. [26] observed the contamination of *L. sativa* when irrigated with water containing microcystin at a concentration of 1700 mg L^−1^ for 10 days; therefore, this work aims to determine the concentration of total microcystins (MC-LR and MCRR) in edible tissue of leafy green plants that were previously irrigated with polluted water and evaluates the potential risk of MCs levels accumulated in these parts to human health.

## 2. Results

### 2.1. Cell Count of Two Dominant Species (Microcystis aeruginosa (Kützing) and Oscillatoria limnetica Lemmermann) and Microcystin Concentration in Irrigation Water

Water samples were collected from irrigation during October. The species composition of phytoplankton recorded in the water are shown in Table 1. *Oscillatoria limnetica* and Microcystis aeruginosa were detected with an average cell count of 2,300,000 and 450,000 cells/mL, respectively, as shown in Table 1. This increase in cell count may be due to the temperature being favorable for cyanobacterial growth (blooming). Dissolved and particulate MCs were detected in the irrigation waters by HPLC with concentrations of 45.04–600 μg/L. MCs in the water samples exceeded the WHO safety limit (1 μg/L) of MC in drinking water. Particulate MCs had positive correlations with the total number of cyanobacteria cells *Oscillatoria limnetica* and Microcystis aeruginosa (r = 0.989–0.998). The dominant and abundant species isolated from irrigation during October were found to produce different amounts and profiles of MCs (Figure 1 and Figure 2 and Table 2). Methanolic extract of the cyanobacteria Oscillatoria limneteca and Microcystis aeruginosa contained mainly MC-LR with minor MC-RR, when compared to MC-LR and MC-RR standard (Figure 1). *Oscillatoria limnetica* could produce two MC variants (MC-LR, RR), these variants showed different proportions, where MC-LR represented the highest proportion (4 × 10^4^ µg/cell) and MC-RR represented the lowest one 1.8 × 10^4^ µg/cell. The intracellular concentration of microcystin from the culture of *Oscillatoria limnetica* was 58,000 μg/L, while it was 7305 μg/L. The particulate concentration of MCs (intracellular MCs) inside the culture of Microcystis aeruginosa was 87,000 μg/L, while dissolved microcystin concentration (extracellular MCs) was 1000 μg/L. Microcystin concentration in each pure culture had a positive correlation with the total number of Microcystis aeruginosa and *Oscillatoria limnetica* (r = 0.507–0.654, *p* = 0.010).

### 2.2. Accumulation of Microcystin-LR and RR in Edible Parts of the Two Vegetable Plants (Eruca sativa and Lactuca sativa L.)

MCs were detected in edible parts of the vegetable plants (*Eruca sativa* and *Lactuca sativa* L.) irrigated with the contaminated waters by HPLC (Table 3). The MCs detected in edible tissues of vegetable plants did not differ greatly among different plants (*p* > 0.05. The level of accumulated microcystin was higher than the limit of the World Health Organization (WHO) (0.04 μg kg^−1^ body weight) for human food consumption. The concentration of MC accumulated in green *L. sativa* irrigated with contaminated water was (1089 ng g^−1^ FW), as shown in Figure 3. This is also higher than the limit of WHO. The concentrations of MCs accumulated in green *L. sativa* was slightly higher than concentrations of MCs accumulated in fresh tissue of *E. sativa*, which was 1044 ng g^−1^ FW and also exceeded the limit of WHO.

### 2.3. Risk Assessment and Potential Health Hazards of Microcystin on Plants (Eruca sativa L. and Lactuca sativa)

To assess the human health risk, the estimated daily intake (EDI) of MCs via consumption of green vegetables that are commonly consumed in Egypt was calculated and compared with the estimated daily intake (EDI) of the World Health Organization limit (0.04 μg MC-LR/kg body weight (BW)/day) [27]. Taking into account that green plants could be consumed in most countries, including Egypt, as fresh green plants and could be consumed daily, then, if a 60 kg adult person consumes 300 g (FW) of green contaminated plants with MCs, 1089 and 1044 ng/g fresh weight as reported in our study (Table 4), the corresponding EDI would be 3630 and 34,801 ng MCs/kg BW/d, respectively. Meanwhile, if 25 kg children consume 100 g of such MC-contaminated plants green beans, EDI values are 4338 and 4176 (ng/kg BW/d) and are higher than those estimated for adults and represent more danger because of less sensitivity in children (Table 4). This indicates that green plants cultivated under these conditions worldwide could represent a risk to human health through food consumption. The toxin concentrations recorded in plants exceeded this limit and could represent a health risk for both adults and children. This indicates that green plants cultivated under these conditions worldwide could represent a risk to human health through food consumption. As a result, the impact of irrigation water contaminated with poisonous cyanobacteria on plants should be closely monitored and MC levels in irrigation waters should be evaluated frequently, particularly in places where cyanobacteria blooms are prevalent. Furthermore, before being delivered into commerce, all plants used for human consumption must be tested for the presence of MCs and other cyanotoxins regularly.

## 3. Discussion

The present study reported the occurrence of microcystin-producing species of cyanobacteria in water that are used for irrigation of crop plants (*Eruca sativa* and *Lactuca sativa* L.) in southern Egypt. Two dominant cyanobacterial species, *Microcystis aeruginosa* and *Oscillatoria limnetica*, were present with high cell density (blooms). HPLC analysis revealed that cyanobacterial cells in naturally occurring blooms of these species produced MCs with varying concentrations and profiles. Generally, the most common MCs variants produced by these species during the present study were MC-LR and -RR. Detection of MC variants in water is important and critical for setting guideline values, as some variants (e.g., MC-RR) are less toxic than other variants, e.g., MCLR [28]. These MC-producing species dominated phytoplankton populations and constituted most cyanobacterial blooms in our sample during autumn. The findings support the notion that cyanobacteria prefer higher temperatures for growth [21]. For example, the highest concentration of particulate MCs in water was linked to the highest cell density of *Oscillatoria limnetica* and *Microcystis aeruginosa*. Other studies have found a link between MC production in cyanobacterial blooms and the cell density of dominant species in the aquatic environment [28]. According to previous research on toxin production in cyanobacteria, MC-LR is the most toxic and commonly found cyanotoxin variant in freshwater sources around the world [29]. It is also a specific inhibitor of the protein phosphate enzymes 1 and 2 A, which regulate a variety of cellular functions in higher plants [30]. In addition to intracellular MCs, dissolved extracellular MCs were discovered in natural waters [28]. MCs were found accumulated in the edible tissue of two crop plants (*Eruca sativa* and *Lactuca sativa* L.) in our study.

Previous research has also revealed that when vegetables and other plants are irrigated with MC-contaminated water, significant amounts of MCs accumulate, posing a risk to human health [31]. Soils growing leafy vegetables had higher MC concentrations than those growing fruit and root vegetables. The concentrations of MCs in soils growing various vegetable species were first linked to the volume of irrigation water used. Water quotas for agriculture in Guangdong and Yunnan provinces of China demonstrated that 1.2–3.3 times more irrigation waters were used for leafy vegetables than for fruit and root vegetables [31]. A great concern about the accumulation of MCs in green beans is the potential human risk from consumption of such contaminated food [32]. MC bioaccumulation affects human health via diet [31,33,34]. The estimated daily intake (EDI) was calculated to assess the health risk of MC-LR from consuming the edible parts (leaves) of the planted vegetables. Adult EDI values of MCs in plants ranged from 3480–3630 ng/kg/d, both of which were 87–90-fold higher than the WHO detection limit (Table 4). The EDI values of MCs in plants for children ranged from 4167 to 4356 ng/kg/d, which were both 104–108 times higher than the WHO detection limit (Table 4). As a result, higher microcystin concentrations in irrigation water may result in greater MC bioaccumulation, as well as increased phytotoxicity and human health risks.

## 4. Materials and Methods

### 4.1. Collection of Water Samples and Isolation of Oscillatoria limnetica and Microcystis aeruginosa

Integrated water samples were collected by cleaned plastic bottles at a depth of about 30 cm from one of the sources of water used to irrigate agricultural lands for crop plants. At the same time, samples from plants *Lactuca sativa* L. and *Eruca sativa* were also collected because they had previously been irrigated with contaminated water. The cyanobacteria *Oscillatoria limnetica* and *Microcystis aeruginosa* were isolated from some water sources used in irrigation of crop plants (*Lactuca sativa* L. and *Eruca sativa*). The organisms were grown separately in a 250 mL conical flask containing BG-11 medium at 30 °C and 50 mol m^−2^ s^−1^ illumination with a 16:8 light/dark period.

The cells were harvested at the exponential stage and centrifuged for 15 min at 6000× *g*. Extracellular MCs were determined from the cell-free supernatant using HPLC. The pellet was extracted in methanol (80%) overnight at room temperature of 25 °C to determine intracellular MCs. The supernatants were blended and the organic solvent was evaporated using sterile air. Toxins in the remaining aqueous fraction were removed using a C-18 cartridge. Toxins were then eluted with 80% methanol, evaporated to dryness, and reconstituted in 1 mL of methanol. Toxin concentrations were determined using Agilant-1200 high-performance liquid chromatography (HPLC) with a UV photodiode-array detector set to 238 nm. Chromatographic separation was carried out on a Zorbax eclips—C18 (150 mm 4.6 mm, 5 m) column (USA), using the conditions previously described by [35]. Extracellular MCs in a cell-free medium of *Oscillatoria limnetica* and *Microcystis aeruginosa* were also detected by HPLC using the same method described above. Dissolved and particulate MCs were detected in the irrigation waters by HPLC.

### 4.2. Determination of Microcystin Accumulated in Plants (Lactuca sativa L. and Eruca sativa)

Shoots (3 g) were collected from each plant, washed three times with distilled water, thinly sliced horizontally, and stored at −80 °C for microcystin analysis. They were homogenized separately in a mortar with 20 mL of 80% methanol. Each homogenate was wrapped in aluminum foil and left at room temperature overnight. Each plant extract was centrifuged at 6000× *g* for 15 min before being re-extracted in 80% methanol. The supernatants were collected and evaporated with sterile air under dry conditions. The dried extracts were completely dissolved in methanol and filtered through a 0.2 m nylon syringe filter. MCs in the supernatant of plants were also detected by HPLC using the same method described above.

### 4.3. Calculation of Estimated Daily Intake

Based on our data, MCs concentrations were detected in (*Lactuca sativa* L. and *Eruca sativa*). Estimated daily vegetable consumption is calculated according to [36].
EDI (μg/kg BW/d) = (MC × DC)/BW

We assumed that adults consume 300 g of *Eruca sativa* and *Lactuca sativa* per day, whereas children consume only 100 g. Where MC is MCs concentration (ng g^−1^ FW), DC is the daily vegetable consumption and BW is the average body weight (adults, 60 kg, and children, 25 kg).

### 4.4. Statistical Analysis

Differences in the total number of cyanobacteria and accumulating MCs in *Eruca sativa* and *Lactuca sativa* were determined by one-way ANOVA. Results were considered significant at *p* < 0.05.

## 5. Conclusions

The presence of toxic *Oscillatoria limnetica* and *Microcystis aeruginosa* and/or their MC toxins in irrigation water poses a risk to human and animal health. MC concentrations in plants (*Eruca sativa* and *Lactuca sativa*) irrigated with water contaminated with cyanobacterial cells led to estimates of daily MCs intake (3480–3630 ng kg^−1^ body weight, respectively) that exceeded the total daily intake guidelines (0.04 μg kg^−1^ body weight) for human food consumption. This study suggests that irrigation water, as well as plants used in human consumption, should be regularly monitored for the presence of MCs and other cyanobacterial toxins.

## Figures and Tables

**Figure 1 plants-11-01733-f001:**
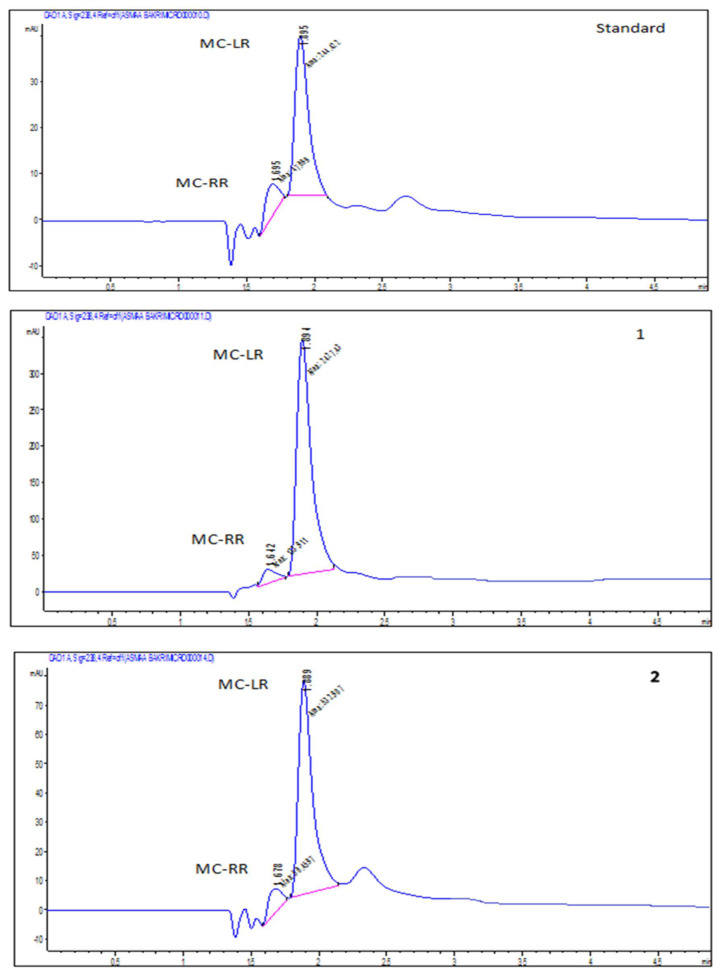
HPLC profile of standard microcystin, extracellular (1) and intracellular (2) MCs produced from *Oscillatoria limnetica Lemmermann* isolated from some of the water sources used in irrigation of crop plants in Sohag district.

**Figure 2 plants-11-01733-f002:**
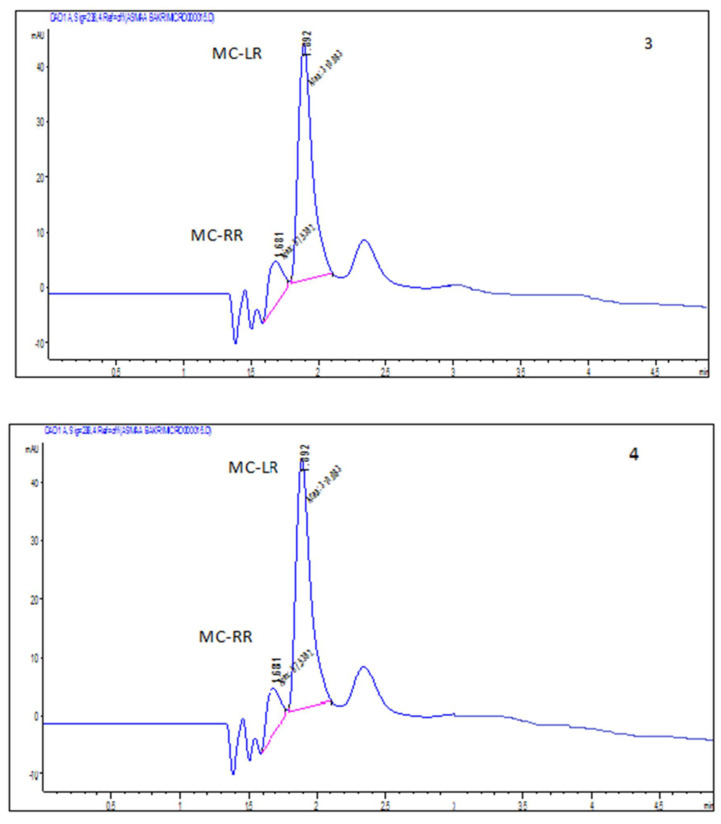
HPLC profile of extracellular (3) and intracellular (4) MCs produced from *Microcystis aeruginosa* isolated from some of the water sources used in irrigation of crop plants in Sohag district.

**Figure 3 plants-11-01733-f003:**
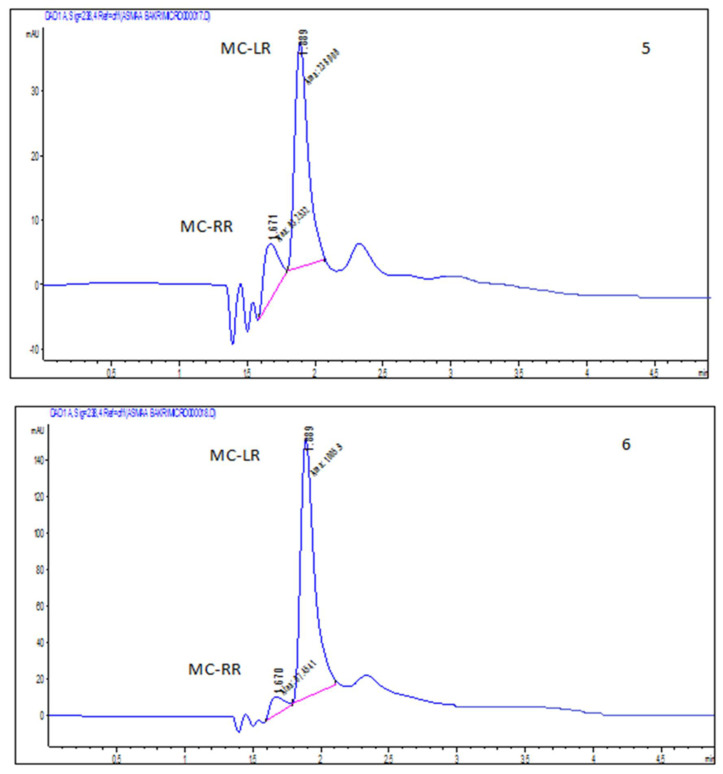
HPLC profile of accumulated MCs from *E. sativa* (5) and *L. sativa* (6) collected from the same site that were previously irrigated with polluted water.

**Table 1 plants-11-01733-t001:** Cell density (cell mL^−1^) of phytoplankton present in water samples used in irrigation of crop plants (*Eruca sativa* and *Lactuca sativa*).

Algal Species	Cell Number (Cells/mL)
Cyanobacteria	
*Oscillatoria limnetica Lemmermann*	2,300,000
*Merismopedia minima G.Beck*	20
*Merismopedia tenuissima Lemmermann*	16
*Microcystis aeruginosa* (*Kützing*)	450,000
*Synechocystis aquatilis Sauvageau*	22
Chlorophyta	
*Ankistrodesmus gracilis* (*Reinsch*)	20
*Chlorella vulgaris Beyerinck*	80
*Chlorococcum lobatum* (*Korshikov*)	88
*Pediastrum duplex Meyen*	96
*Scenedesmus ellipsoideus Chodat*	10
*Ankistrodesmus gracilis* (*Reinsch*)	20
*Chlorella vulgaris Beyerinck*	80
*Chlorococcum lobatum* (*Korshikov*)	88
*Pediastrum duplex Meyen*	96
Bacillariophyta	
*Cyclotella* sp.	5
*Fragillaria* sp.	12
*Melosira* sp.	8
*Navicula* sp.	6
*Nitzschia* sp.	4
*Tribonema* sp.	4

**Table 2 plants-11-01733-t002:** Concentrations of intracellular and extracellular microcystins in *Microcystis aeruginosa* and *Oscillatoria limnetica* isolated from water.

Samples	MC-RR (μg/L)	MC-LR (μg/L)	Total (μg/L)
Intracellular MCs (*Oscillatoria limnetica)*	18,000	40,000	58,000
Extracellular MCs (*Oscillatoria limnetica*)	2300	5005	7305
Intracellular MCs of (*Microcystis aeruginosa*)	1000	87,000	88,000
Extracellular MCs (*Microcystis aeruginosa*)	20	285	305

**Table 3 plants-11-01733-t003:** Concentrations of microcystin (MC-LR and MC-RR) accumulated in edible parts of two plants (*Eruca sativa* and *Lactuca sativa* L.).

	MC-RR (ng g^−1^ FW)	MC-LR (ng g^−1^ FW)	Total (ng g^−1^ FW)
MCs accumulated *E. sativa*	89	1000	1089
MCs accumulated *L. sativa*	56	988	1044

**Table 4 plants-11-01733-t004:** Estimated daily MCs intake through food consumption of edible parts of two plants. (μg/kg/d).

	EDI (Adults) ng/g FW	EDI (Children) ng/g FW
MCs accumulated *E. sativa*	3630	4356
MCs accumulated *L. sativa*	3480	4176

## Data Availability

The datasets generated and/or analyzed during the current study are available from the corresponding authors upon reasonable request.

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
