# Peer review of "Accumulation of Microcystin from Oscillatoria limnetica Lemmermann and Microcystis aeruginosa (Kützing) in Two Leafy Green Vegetable Crop Plants Lactuca sativa L. and Eruca sativa"

_plants, 2022, doi:10.3390/plants11131733_

Round 1

Reviewer 1 Report

The revision is much better. 

Author Response

Thank you very much for your critical and helpful comments on many grammatical mistakes we made in the original MS. We carefully checked all these grammatical mistakes and revised as possible as we could by using 

Grammarly: Free Online Writing Assistant

https://app.grammarly.com/

Thank you so much 

Reviewer 2 Report

The authors have carefully revised this manuscript, but moderate English changes required.

Author Response

Thank you very much for your critical and helpful comments on many grammatical mistakes we made in the original MS. We carefully checked all these grammatical mistakes and revised as possible as we could by using 

Grammarly: Free Online Writing Assistant

https://app.grammarly.com/

Thank you so much 

This manuscript is a resubmission of an earlier submission. The following is a list of the peer review reports and author responses from that submission.

Round 1

Reviewer 1 Report

This manuscript is about a very important research topic. However, the manuscript lack of writing quality meaning that it has many mistakes and is confusing; scientific names of cyanobacteria poorly writing; througouth the text autors refer to green vegetable crops and does not mention the actual names of the crops used; tables with duplicated results, and son on. It seems that the authors just include some sentences on the manuscript ad did not care abou to read the manucript.

Reviewer 2 Report

This manuscript reported the concentrations of MC-LR and MR-RR in two microcystin-produced algaes and two leafy vegetables irrigated with water containing the two algaes in Egypt, showing high health risks by eating the vegetables. The finding of this study is some of significance in understanding the pollution effect raised by the microcystins in soil-plant system. Some comments are listed for authors improving the manuscript as follows:

(1) Please carefully cheak the manuscript, since there are many format and grammar errors. For example, why not use the  complete "()"?

(2) Please improve the figures resolution ratio and clearly clarify which are the MC-LR and MC-RR in the  chromatograms.

(3) Please explain why just analyze the two microcystins, .i.e., MC-LR and MC-RR? 

(4) The data size in this manuscript is limited. More detection of the microcystins for various crops irrigated with the microcystins-containing water in Egypt are encouraged. 

(5) Please clearly clarify which and how the vegetables were sampled in this study.

(6) Please provide the data on recoveries and detection limit of the target MCs.